# Post-polymerisation functionalisation of conjugated polymer backbones and its application in multi-functional emissive nanoparticles

Adam Creamer[1,2], Christopher S. Wood[3,4,5], Philip D. Howes[3,4,5,6], Abby Casey[1,2], Shengyu Cong[1,2], Adam V. Marsh[1,2], Robert Godin [1,2], Julianna Panidi[2,7], Thomas D. Anthopoulos [7,8], Claire H. Burgess[3], Tingman Wu[1,2], Zhuping Fei[1,2], Iain Hamilton[2,7], Martyn A. McLachlan[3], Molly M. Stevens [3,4,5] & Martin Heeney [1,2]

Backbone functionalisation of conjugated polymers is crucial to their performance in many applications, from electronic displays to nanoparticle biosensors, yet there are limited approaches to introduce functionality. To address this challenge we have developed a method for the direct modification of the aromatic backbone of a conjugated polymer, post-polymerisation. This is achieved via a quantitative nucleophilic aromatic substitution ($S_NAr$) reaction on a range of fluorinated electron-deficient comonomers. The method allows for facile tuning of the physical and optoelectronic properties within a batch of consistent molecular weight and dispersity. It also enables the introduction of multiple different functional groups onto the polymer backbone in a controlled manner. To demonstrate the versatility of this reaction, we designed and synthesised a range of emissive poly(9,9-dioctylfluorene-*alt*-benzothiadiazole) (F8BT)-based polymers for the creation of mono and multifunctional semiconducting polymer nanoparticles (SPNs) capable of two orthogonal bioconjugation reactions on the same surface.

[1] Department of Chemistry, Imperial College London, London SW7 2AZ, UK. [2] Centre for Plastic Electronics, Imperial College London, London SW7 2AZ, UK. [3] Department of Materials, Imperial College London, London SW7 2AZ, UK. [4] Department of Bioengineering, Imperial College London, London SW7 2AZ, UK. [5] Institute of Biomedical Engineering, Imperial College London, London SW7 2AZ, UK. [6] Institute for Chemical and Bioengineering, Department of Chemistry and Applied Biosciences, ETH Zürich, Zürich 8093, Switzerland. [7] Department of Physics, Imperial College London, London SW7 2AZ, UK. [8] Physical Sciences and Engineering Division (PSE) King Abdullah University of Science and Technology (KAUST), Thuwal 23955-6900, Saudi Arabia. These authors contributed equally: Adam Creamer, Christopher S. Wood. Correspondence and requests for materials should be addressed to M.M.S. (email: m.stevens@imperial.ac.uk) or to M.H. (email: m.heeney@imperial.ac.uk)

Conjugated polymers are a highly promising class of material for a diverse range of applications, from the active components of optoelectronic devices[1] to semiconducting polymer nanoparticles (SPNs)[2–4]. The key to their application in these areas is the ability to tune both the electronic and physical properties of the polymer and introduce functional groups onto the backbone[5]. To date most approaches to manipulate such properties have relied on the copolymerisation of appropriate monomeric materials[6]. For example, the incorporation of varying amounts of electron-deficient monomers into an electron-rich polymer backbone has a significant impact on the optical band gap due to the hybridisation of the frontier molecular orbitals. Similarly the polymer crystallinity and processability can be tuned by inclusion of monomers with branched or linear sidechains. However, the establishment of structure–property relationships in such copolymers is complicated by the necessity to synthesise batches of varying comonomer ratios, which is both time consuming and challenging to control. Therefore, it is often difficult to disentangle the influence of changes in the backbone chemistry from variations in the molecular weight, end-groups and dispersities of different batches[7].

In principle, the above-mentioned difficulties could be overcome by direct modification of the aromatic backbone of the conjugated polymer post-polymerisation, thus enabling the synthesis of libraries with consistent dispersity and length. In order to have a practical significance, any such reaction on a polymer backbone should be both highly selective and quantitative in yield[8]. Click reactions fit these criteria well[9], and have had an important and growing influence in the field of materials science over the last decade[10,11]. Many of the common click reactions have also been applied to conjugated polymers[12], but almost exclusively these have involved functional groups that are decoupled from the aromatic backbone via a saturated alkyl spacer, and thus have limited influence on the electronic properties of the polymer. Post-polymerisation reactions directly on the aromatic backbone itself have rarely been reported, mainly due to issues with selectivity and incomplete reactions. Those that have been reported utilise unusual building blocks like dibenzocyclooctyne[13] or very specific chemistries like benzannulation[14].

Furthermore, it is often desirable to introduce functional groups onto the backbone. For example, groups such as acrylates, carboxylic acids, azides, alkynes and so-on have been added to facilitate self-assembly, to enable crosslinking and photopatterning or to promote interactions with metal oxide interfaces. Typically such reactive groups are not able to survive the polymerisation conditions and are introduced via functional group interconversions after polymerisation. In many cases it would be desirable to have two or more different functional groups on the same polymer backbone. For example two orthonal click groups like an azide and an alkene could be used for crosslinking and subsequent surface functionalisation or for dual sensing applications. The preparation of such conjugated polymers via current methodologies is complex, and as such they are rarely reported[15]. Direct backbone functionalisation offers an attractive method for such functionalisation. This is of particular interest in the field of SPNs, which are relatively challenging to functionalise. Existing strategies to SPN functionalisation include encapsulation by an inorganic shell and subsequent chemical modification[16–18], or coprecipitation/emulsification with amphiphilic molecules containing reactive groups[19–21]. This later approach relies on non-covalent interactions between the polymer and the functionalised counterpart, which can dissociate over time[22], particularly at elevated temperature. All of these approaches also rely on dilution of the emissive component, which inherently limits the brightness of the particle for a given size. Covalent attachment of functional groups directly to the semiconducting polymer backbone overcomes these problems[23], but has been synthetically challenging, particularly if multiple functional groups are required.

Here, we report a widely applicable strategy to directly modify the conjugated backbone via a quantitative nucleophilic aromatic substitution ($S_NAr$) reaction on fluorinated electron-deficient comonomers within the polymer backbone (Fig. 1a). Such functionalisation results in significant changes to the optoelectronic and physical properties of the polymer. It also enables the facile introduction of reactive handles for further functionalisation or cross-linking. Multiple reactive groups can be introduced in a controlled one-pot reaction. Our approach to synthesising these materials enables the facile creation of functionalised polymers and particles with multiple different reactive handles covalently linked on their surface.

## Results

**Post-polymerisation reaction.** We first investigated whether the post-polymerisation substitution reaction of a polymer containing a fluorinated acceptor (5-fluorobenzo-2,1,3-thiadiazole, FBT) could be performed without undesirable side reactions on the polymer backbone. Initial experiments focused upon displacement reactions with alkanethiols due to their high nucleophilicity[24]. Thus treatment of the carbazole copolymer **P1-F** with an excess of octanethiol in the presence of base in a mixed solvent of chlorobenzene/DMF was found to result in complete substitution of the fluoride after 30 min at 120 °C, affording **P1-SR** (Fig. 1b).

**Fig. 1** General reaction scheme and example reaction. **a** Scheme for the modification of semiconducting polymers containing either fluorinated benzotriazole or benzothiadiazole comonomers with a thiol or alcohol. **b** Synthesis of P1-SR from two different strategies: (i) 9-(9-heptadecanyl)−9H-carbazole-2,7-diboronic acid bis(pinacol) ester, Pd(PPh₃)₄, toluene, Na₂CO₃ (aq), 120 °C 3 days. (ii) 120 °C 30 min (microwave), excess Na₂CO₃, 3:1 (v:v) Chlorobenzene (CB):DMF

This polymer was found to have near identical UV–Vis absorption and [1]H NMR spectra to a comparative polymer **P1-SR-Suz**, synthesised via the traditional Suzuki copolymerisation of the two appropriate comonomers[25] (Supplementary Figures 1 and 2). [19]F NMR spectroscopy on **P1-SR** indicated a complete absence of fluorine signals demonstrating that the $S_NAr$ reaction can succesfully substitute all of the backbone fluorine groups, without adversely affecting other parts of the polymer backbone (Supplementary Figure 3). Given that the performance of semiconducting polymers in electronic devices like organic thin-film transistors (OTFTs) is very sensitive to the presence of chemical defects or impurities, the performance of both polymers was also investigated in bottom gate, bottom contact OTFTs. Gratifyingly both polymers exhibited similar overall performance, with comparable charge carrier mobility (see Supplementary Figure 4 and Table 1), providing further evidence that the displacement chemistry does not adversely affect polymer performance.

To further probe the utility of the reaction, we moved to a more soluble polymer, poly(9,9′-dioctylfluorene-5-fluoro-2,1,3-benzothiadiazole) (**P2-F**), which could be readily prepared at a higher molecular weight ($M_n \approx 113 \text{ kg mol}^{-1}$) than the carbazole copolymer (**P1-F**) ($M_n \approx 4 \text{ kg mol}^{-1}$). The non-fluorinated analogue is a highly emissive commercially available material commonly used in SPNs[4,26]. In order to test if the alkanethiol could be quantitatively incorporated onto the polymer backbone, we performed five separate reactions with increasing equivalents of 1-octanethiol nucleophile (calculated vs. the $M_w$ of the repeat unit) under similar reaction conditions to those for the carbazole

copolymer (Fig. 2). Following work-up, the [1]H NMR spectra demonstrated fluoride displacement, with the –$SCH_2$– protons signals clearly observable ($\delta$ 3.03 ppm) in a region of the spectra with no existing signals (Fig. 2c). Integration of these new signals against the aromatic region (7H) established that quantitative displacement had occurred. Furthermore, combustion analysis of the fluorinated and fully substituted polymer showed good agreement with the predicted values (see Supplementary Methods). The reaction is further notable for the high $M_w$ starting polymer used, demonstrating that the backbone is still accessible for reaction. However the high $M_w$ did complicate analysis of the reacted polymers by matrix-assisted laser desorption/ionisation time of flight (MALDI-TOF) mass spectroscopy due to the poor volatility of the samples. Therefore, complete displacement was also verified on a lower weight polymer via MALDI (see Supplementary Figure 5).

The UV–Vis absorption spectra of the polymers as a function of thioalkyl substitution are shown in Fig. 2b. Thioalkyl substitution did not have a significant impact on the absorption edge, but increasing substitution does reduce the relative intensity of the long wavelength intramolecular charge transfer (ICT) band compared to the high-energy absorption around 330 nm. A new absorption peak is also observed around 370 nm, which increases in intensity as function of thioalkyl substitution. The photoluminescence (PL) spectra do not signficantly change as function of substitution (Supplementary Figure 6). The PL quantum yield (PLQY) was found to reduce from ca. 89% for the fully fluorinated polymer to 66% for the fully substituted polymer in

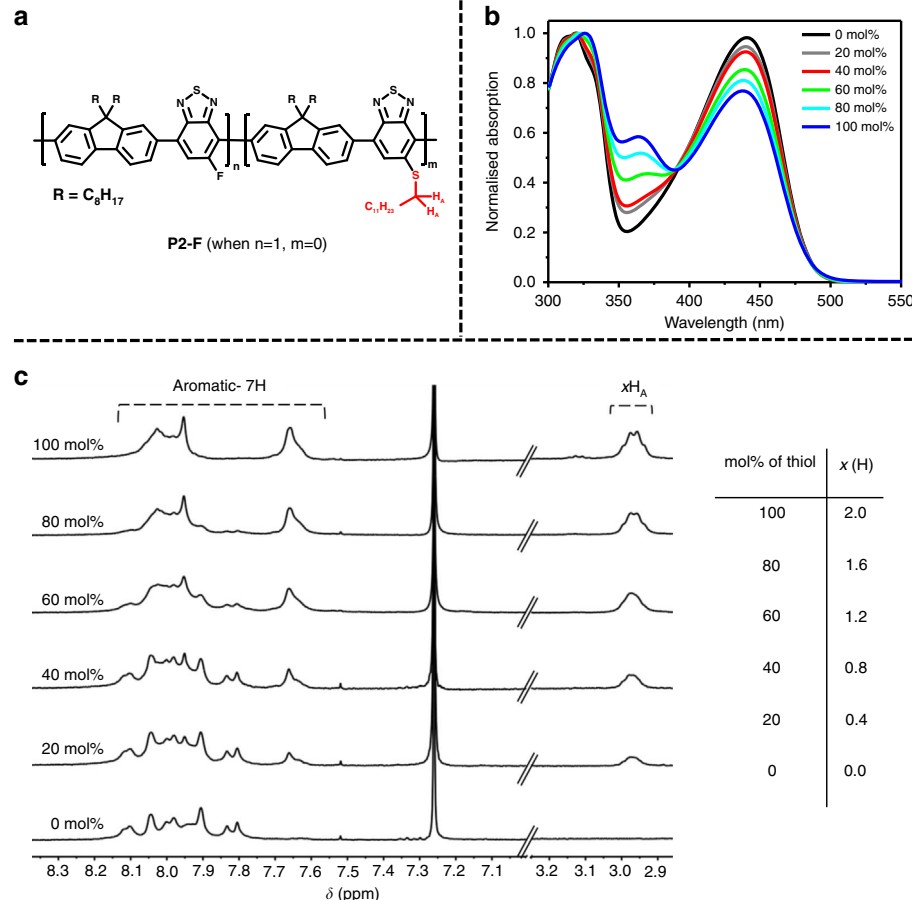

**Fig. 2** Control of dodecanethiol substitution onto the backbone. **a** Structure of substituted polymer **P2-F**. **b** Normalised UV–Vis absorption spectra of each polymer, in chlorobenzene solution. **c** Stacked [1]H NMR spectra with increasing mol% of thiol, aromatic peaks integrated to 7H for each spectra. Integration value for $H_A$ ($x$) listed in table for each mol% of thiol reacted

THF solution. It should be noted that upon formation of polymer nanoparticles in water (vide infra) the substitution reaction had little impact on the PLQY (from 47% to 40%) (Supplementary Figure 7).

Having established the utility of the reaction with simple alkanethiols, we investigated the versatility of the post-polymerisation reaction with a selection of functionalised thiols, thioacetates and alcohols, using **P2-F** as our workhorse material due to its high solubility (Fig. 3a). We were particularly interested in using this methodology to incorporate functional sidechains amenable to further reaction under mild conditions, i.e. azides, carboxylic acids, thiols and alkenes. The ability to prepare such functionalised materials in one step with good control of the substitution level is desirable for many applications[10].

The chemical structures of all polymers are depicted in Fig. 3a. The methods section contains the details of their synthesis, and the Supplementary Information contains $^1H$ NMR (Supplementary Figures 8–16) and UV–Vis absorption spectra

(Supplementary Figures 17 and 18). Where applicable, $^{19}F$ NMR (Supplementary Figure 19) and IR spectra (Supplementary Figure 20) are also included. Azide and alkene terminated polymers were synthesised via protected thioacetates (*S*-(3-azidopropyl)thioacetate and *S*-(10-undecenyl)thioacetate respectively), due to the relatively poor stability of the free thiols. An in situ deprotection of the thioacetate by treatment with NaOH in the mixed chlorobenzene (CB)/DMF solvent was performed and full substitution onto the polymer backbone was achieved after 10 min at 100 °C. In the case of carboxylic acid and thioacetate functionalised polymers, substitution of all fluorine atoms resulted in an insoluble polymer. However, fully soluble polymers **P2-COOH** and **P2-SAc** could be achieved by reducing loading of the functional thioalkyl group to 25 and 35 mol%, respectively.

The substitution reaction was also found to be successful with the less nucleophilic alcohol group. Treatment of **P2-F** with 2-ethylhexanol or triethylene glycol monomethyl in a 3:1 (v:v) mix of chlorobenzene and DMF in the presence of excess KOH

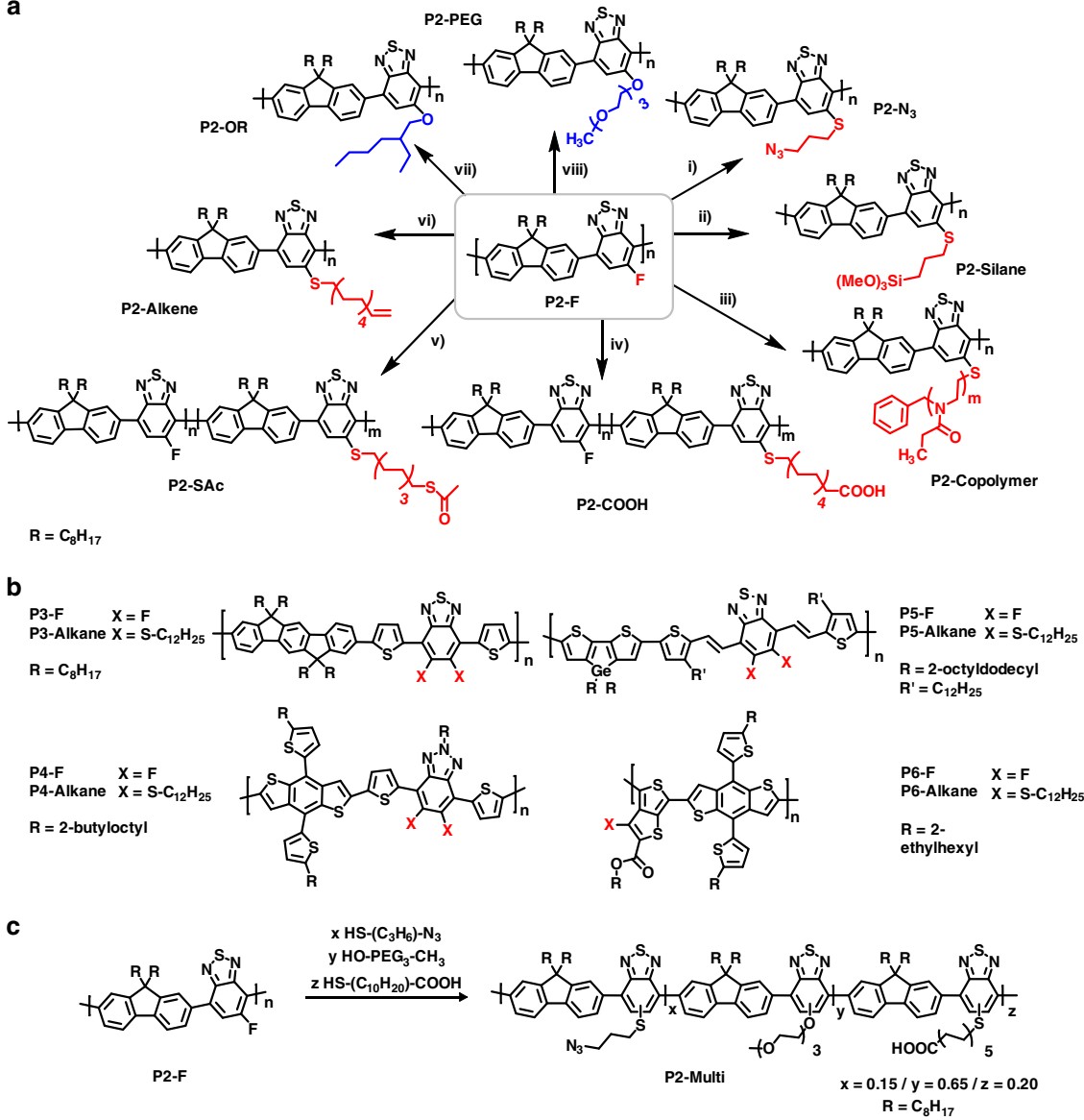

**Fig. 3** Scope of the reaction. **a** Reaction of **P2-F** with functionalised thiols, thioacetates and alcohols: all reactions in CB:DMF (3:1, v-v) solvent. (i) *S*-(3-azidopropyl)thioacetate, KOH; (ii) (3-mercaptopropyl)trimethoxysilane, $K_2CO_3$; (iii) poly(2-ethyl-2-oxazoline), $K_2CO_3$; (iv) 11-mercaptoundecanoic acid, $K_2CO_3$; (v) 1,8-octanedithiol diacetate, NaOH; (vi) *S*-(10-undecenyl)thioacetate, KOH; (vii) 2-ethyl-1-hexanol, KOH; (viii) triethylene glycol monomethyl ether, KOH. **b** Structures of polymers **P3–P6** before and after thiol substitution. **c** Schematic of the synthesis of the multifunctionalised polymer (**P7-Multi**)

yielded polymers **P2-OR** and **P2-PEG**, respectively. The resulting polymers exhibited a slightly different UV–Vis spectrum to the thiol analogue, with a similar drop in the ICT intensity but with a small red shift, when compared to the fluorinated precursor. However, they did not exhibit the new band at 370 nm which was seen in all thiol-containing polymers.

This approach can even be used to incorporate water sensitive cross-linkable sidechains, such as trimethoxysilanes. Although such groups are commonly used surface modifiers and cross-linkers in commercial polymers they have been rarely utilised as functional groups in conjugated systems[27], likely due to their incompatibility with typical polymerisation conditions. Here though we could readily incorporate such groups by treatment with an excess of (3-mercaptopropyl)trimethoxysilane under anhydrous conditions. The resulting polymer (**P2-Silane**) could be cross-linked into intractable films simply by exposure to ambient air (Supplementary Figure 21), a very mild crosslinking condition which could be beneficial for a number of device applications[28].

We also investigated the synthesis of a graft copolymer. Due to the reduced reactivity of polymer chain ends, coupled with the typically poor reactivity of polymer backbones, this can be considered as the 'gold standard' in terms of establishing reaction utility. Gratifyingly, we found that complete backbone substitution occurred upon reaction with a thiol terminated poly(2-ethyl-2-oxazoline) ($M_n$ ca. 2000 g mol$^{-1}$) after 30 min under our standard conditions. Unlike the starting **P2-F** the resulting graft copolymer (**P2-Copolymer**) showed excellent solubility in methanol as a result of the polar poly(oxazoline) graft.

The substitution reaction also occurs on a range of conjugated polymers containing other fluorinated acceptor comonomers (Fig. 3b). Copolymers of 5,6-difluorobenzothiadiazole with indenofluoroene (**P3-F**) were shown to be amenable to substitution of both of the fluoro groups upon treatment with excess thiol. Polymers containing fluorinated benzotriazole groups (**P4-F**) were also quantitatively substituted, despite the reduced electron accepting nature of benzotriazole compared to FBT. In both of these examples substitution was shown to result in a significant blue shift of the absorption maxima (Supplementary Figure 22a and b), due to a combination of backbone twisting upon inclusion of the larger thioalkyl group and a reduction in the acceptor strength due to the fluoride substitution[25,29]. The backbone twisting could be alleviated by inserting a vinylene spacer group between the BT unit and the comonomer (**P5-F**). Upon substitution, little change in the UV–Vis spectrum was observed (Supplementary Figure 22c). Notably no side reactions were observed between the thiol and the vinylene groups. Fluorinated thieno[3,4-b]thiophene is a prevalent acceptor in many high performance solar cell-based polymers[30], and is also amenable to complete substitution of the fluoride under these conditions (**P6-F**, Supplementary Figure 22d). In all cases substitution was verified by a combination of $^1$H NMR (Supplementary Figures 23–27)and the absence of fluorine signals in $^{19}$F NMR (Supplementary Figures 28–31).

To fully demonstrate the utility of our approach, we prepared multi-functionalised polymers containing carboxylic acids, azides and PEG groups on the same polymer backbone (**P2-Multi**, Fig. 3c). The preparation of such copolymers via conventional routes has, to the best of our knowledge, not been reported and would involve a complex copolymerisation of carboxylic acid and PEG-containing monomers together with precursors that could be selectively converted to azides. In contrast, our preparation was readily achieved by treatment of **P2-F** with 11-mercaptoundecanoic acid (25 mol%) using the same conditions as used in the synthesis of **P2-COOH**. After work-up the polymer was found to contain 20% carboxylic acid groups by $^1$H NMR.

Subsequent reaction with 60 mol% of triethylene glycol mono-methyl ether (HO-PEG$_3$-OMe) in the presence of KOH for 10 min, followed by the addition of excess S-(3-azidopropyl) thioacetate, afforded the polymer containing ~65% ethylene glycol chains, 15% azide and 20% carboxylic acid groups after work-up (as determined by $^1$H NMR, Supplementary Figure 32). Progress of the reaction was also tracked by recording the absorption spectra after addition of each reagent (Supplementary Figure 33). **P7-Multi** with similar ratios could also be synthesised via a one-pot protocol, starting from **P2-F**, in which each nucleophile was simply added sequentially (see Supplementary Figure 34 for NMR).

**Surface functionalised SPNs**. The control afforded from this approach has particular utility in the creation of SPNs, especially with regards to the creation of particles with multifuctional surface chemistries. The conjugation of biomolecules to the surface of SPNs is often necessary for biological applications and many strategies exist to achieve this[17,19,31–35]. Some of the most common approaches involve covalent attachment via azide or carboxylic acid groups present on the nanoparticle surface. We looked to demonstrate that our approach to functionalised semiconducting polymers could be used to create SPNs that can be functionalised by strain-promoted alkyne-azide cycloaddition (SPAAC) reactions and amide bond forming reactions, both separately and on the same particle.

In the present case, azide nanoparticles **SPN-N$_3$** were readily synthesised (from **P2-N$_3$**) using the nanoprecipitation method[36]. The size distribution of resulting nanoparticles was analysed by dynamic light scattering (DLS), scanning transmission electron microscopy (STEM) and nanoparticle tracking analysis (NTA) (Fig. 4b–d) and were ~100 nm in diameter (see Supplementary Figure 35 for size distributions from NTA and STEM analysis). This approach is not limited to azide and other polymers including **P2-Silane** and **P2-COOH** can be used to create particles bearing other functional groups. In the case of **SPN-Silane** nanoparticles functional group incorporation can be confirmed by energy-dispersive X-ray (EDX) spectroscopy, which when combined with STEM imaging can be used to probe and map spatial variation in elemental composition, revealing elevated concentrations of Si in our **SPN-Silane** nanoparticles (Fig. 4e). The nanoparticles also showed sizes similar to **SPN-N$_3$** under DLS analysis (Fig. 4b).

Next, we demonstrated that the functional groups on the polymer were present and free to react at the surface of the particles. We initially investigated the use of the azide-containing nanoparticles, labelled **SPN-N$_3$**. Azides undergo click cycloaddition reactions with alkynes (copper-catalysed) or strained alkynes (copper-free) and are extensively utilised for bioconjugation[10]. To determine if the azides on the surface are free to react, a Förster resonance energy transfer (FRET)-based assay was used, as efficient FRET transfer should only occur when the dye is closely bound to the surface of the nanoparticle. In this assay the reactivity between **SPN-N$_3$** and a strained alkyne bearing a rhodamine-based dye (DBCO-MB 594, Fig. 5a) was investigated, as it exhibited complimentary absorption and fluorescence to the F8BT-based nanoparticle (Fig. 5b). An unreactive dye analogue was synthesised, as a control, by reacting a solution of DBCO-MB 594 with an excess of azidoacetic acid. Both reactive and unreactive dye solutions were then made to the same concentration. The concentration of nanoparticles was calculated via NTA, and the amount of DBCO-MB 594 was gradually increased from 423 equivalents of dye molecules per nanoparticle (given the value of $x$) from $x$ to $16x$ ($x$, $2x$, $4x$, $8x$, $12x$ and $16x$ equivalents). PL spectra of the resulting solutions with excitation at 450 nm

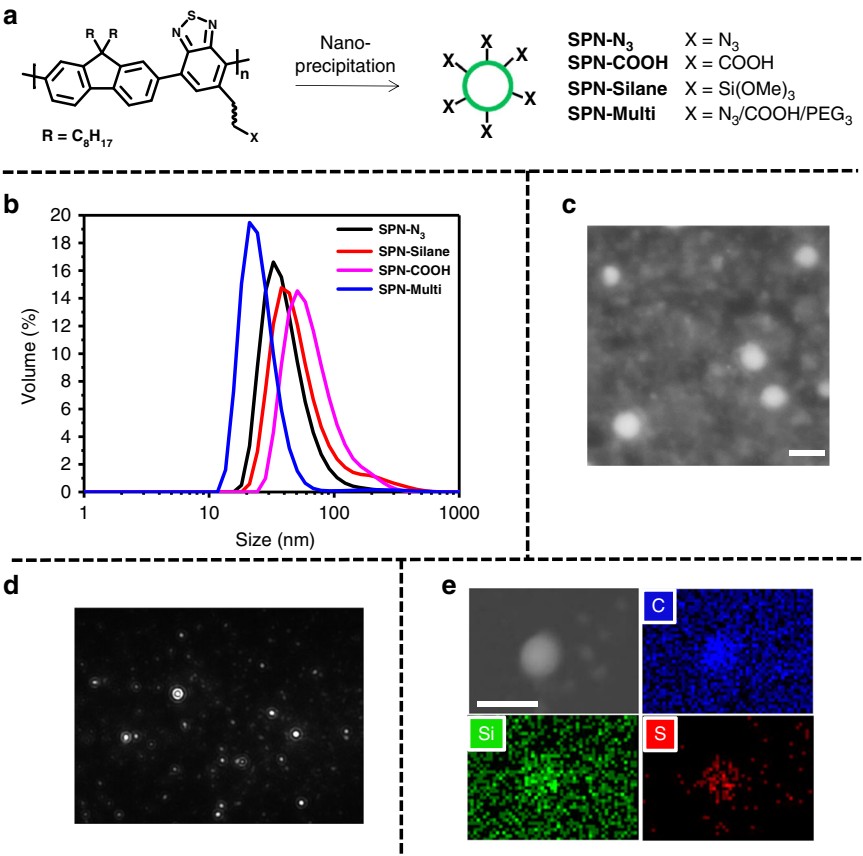

**Fig. 4** SPN synthesis and characterisation. **a** Schematic of the preparation of SPNs from functionalised polymers. **b** Size distribution for **SPN-N₃**, **SPN-Silane**, **SPN-COOH** and **SPN-Multi**, determined by dynamic light scattering (DLS), averaged over three measurements. **c** Scanning transmission electron microscopy (STEM) image of **SPN-N₃** nanoparticles. Scale bar 200 nm. **d** Image of **SPN-N₃** nanoparticle suspension from nanoparticle tracking analysis (NTA). **e** STEM image of **SPN-Silane** nanoparticles, with EDX maps of carbon, silicon and sulphur content. Scale bars, 200 nm

($\lambda_{max}$ of the nanoparticle) are shown in Fig. 5c. The nanoparticles that were combined with the active dye exhibit a significant quenching of the nanoparticle (donor) and enhancement of the dye (acceptor) fluorescence intensity with increasing dye equivalents. However, the analogous reaction with the inactive dye showed a comparatively insignificant donor quenching and acceptor enhancement (Supplementary Figure 36).

FRET efficiencies were calculated from the PL spectra and in the case of active dye, the efficiency was found to plateau at 8x equivalents as the value approaches the maximal FRET efficiency (see Supplementary Discussion for full report on how FRET efficiencies were calculated from PL data). However, in the case of the inactive dye, saturation of the $E_{FRET}$ was not observed since acceptors free in solution can be too far from donor NPs for efficient FRET to occur. The lifetime of the excited state of the donor of the solutions was also measured using time-correlated single photon counting (TCSPC). In both cases substantial differences were seen between the active and inactive dye reactions, consistent with the formation of a covalent linkage (see Supplementary Discussion for report of TCSPC data). Further control experiments, with particles that did not bear azide functional groups, also did not exhibit FRET under similar conditions further indicating that covalent attachment is responsible for the observed effect and that non-specific binding plays an insignificant role (Supplementary Figure 37).

Amide bond forming reactions are another extensively used bioconjugation technique and there are many examples of carboxylic acid-coated nanoparticles, including SPNs, that have been functionalised with amines via this method[33,37,38]. In our case, **SPN-COOH** nanoparticles could be functionalised via the widely used amide coupling reagent 1-ethyl-3-(3-dimethylami-nopropyl)carbodiimide (EDC) and the surface functionalisation reaction assessed through a lateral flow test strip assay. The **SPN-COOH** nanoparticles were reacted with EDC and *N*-hydroxysulfosuccinimide (sulfo-NHS) with varying amounts of a water soluble amine functionalised with biotin *O*-(2-aminoethyl)-*O'*-[2-(biotinylamino)ethyl]octaethylene glycol (biotin-NH₂) (structure in Fig. 5d). This yielded nanoparticles with differing amounts of biotin conjugated to the surface. To test the functionalisation, each resulting mixture was wicked up a nitrocellulose-based lateral flow strip with a test line of adsorbed streptavidin proteins 4 cm from the base. The extremely high binding constant between biotin and streptatividin[39] causes the fluorescent particles to bind at the test line if biotin is present on the nanoparticle surface (Fig. 5e). In this case, the greatest fluorescence intensity (observed under UV irradiation (365 nm)) at the test line was observed for particles reacted with 10 µg mL⁻¹ solutions of PEG–biotin. In all cases no binding was observed in control reactions where EDC was not present (Fig. 5e), providing strong evidence that the carboxylic acid functionalised SPNs can be activated with EDC and react to form amides with suitable amines.

**Multi-functionalised SPNs.** The unique control afforded by the synthesis of multi-functional polymers offers the exciting possibility of creating fluorescent nanoparticles that can be functionalised with multiple different groups via orthogonal coupling reactions. **SPN-Multi** was prepared using the same

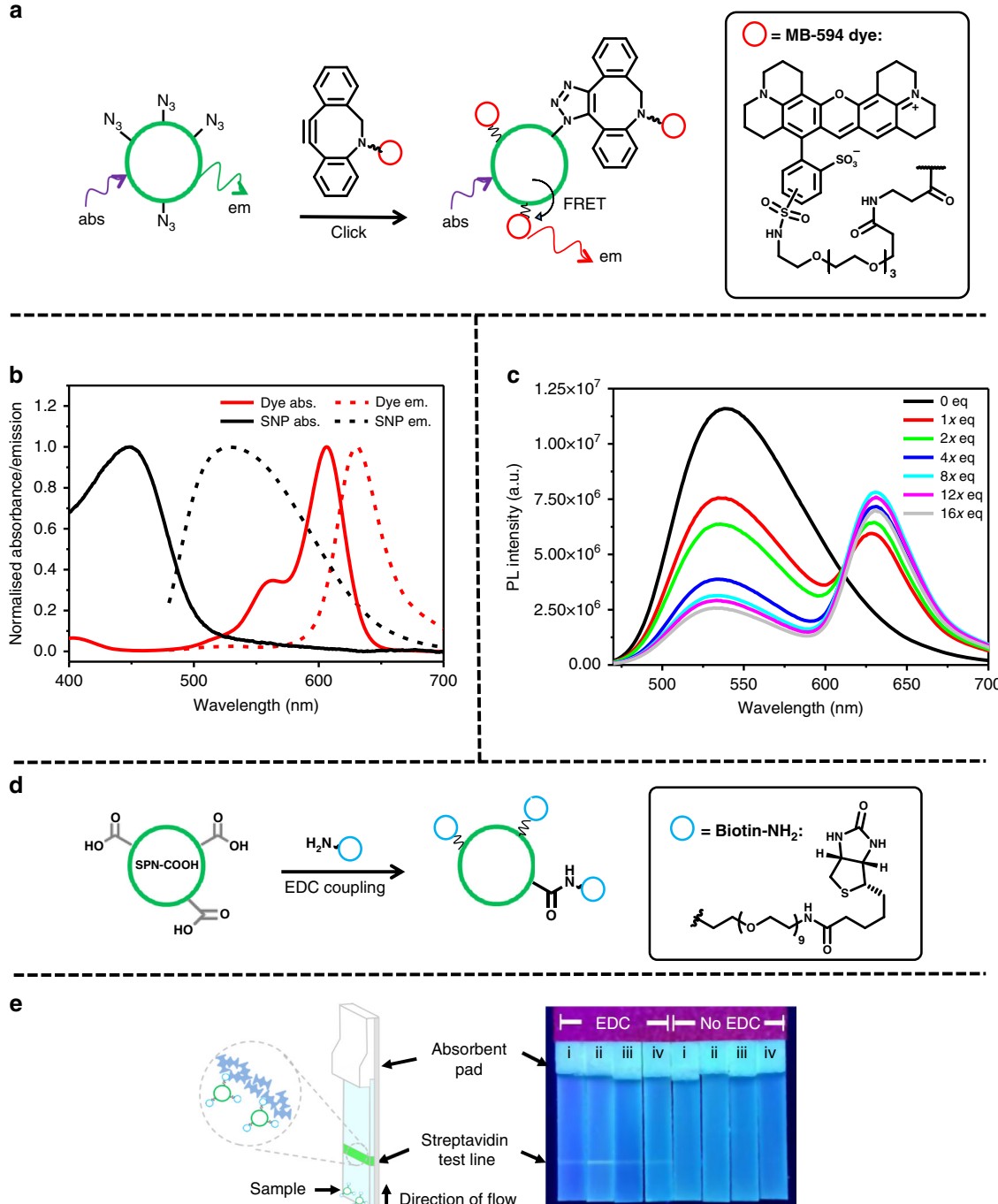

**Fig. 5** SPN surface monofunctionalisation. **a** Schematic of the modification of the surface of **SPN-N₃** with active dye, enabling FRET between dye and nanoparticle. **b** Normalised absorbance (solid lines) and emission (black lines) spectra of the dye (red lines) and the nanoparticles (black lines). **c** PL spectra of **SPN-N₃** with increasing equivalents of active dye, x = 423 dye molecules per nanoparticle (concentration of nanoparticle calculated with NTA) excited at 450 nm, all at the same nanoparticle concentration. **d** Schematic of the EDC coupling on the surface of **SPN-COOH** with a biotin functionalised amine. **e** Illustration of the components of a poly-streptavidin test strip with image of the test line run with particles functionalised with and without EDC present at increasing concentrations of biotin-NH₂: (i) 1 μg mL⁻¹, (ii) 10 μg mL⁻¹, (iii) 100 μg mL⁻¹ and (iv) 200 μg mL⁻¹ in water

nanoprecipitation method as before and particles were found to react with biotin-NH₂ and DBCO-594 separately, in the same fashion as **SPN-COOH** and **SPN-N₃**, respectively (Supplementary Figures 38 and 39). Adding DBCO dye first or at the same time as other reactants was found to inhibit EDC coupling, most likely due to surface bound dye blocking further reactions. However, allowing the EDC coupling of biotin to occur (with 10 μg mL⁻¹ biotin–NH₂), then adding DBCO dye gave rise to the characteristic quenching of the donor (nanoparticle) and

enhancement of the acceptor (dye) in PL spectra (Fig. 6b). Running this solution along a lateral flow strip gave rise to a red test line under UV irradiation (365 nm) (Fig. 6c). This indicated that particles, functionalised with both biotin and dye, were binding at the test line. In contrast, control solutions with unreactive dye (DBCO-MB 594 pre-reacted with an excess of azidoacetic acid) treated in the same way gave PL spectra characteristic of non-functionalised particles, and a correspondingly yellow coloured test line (Fig. 6c). This result could also be

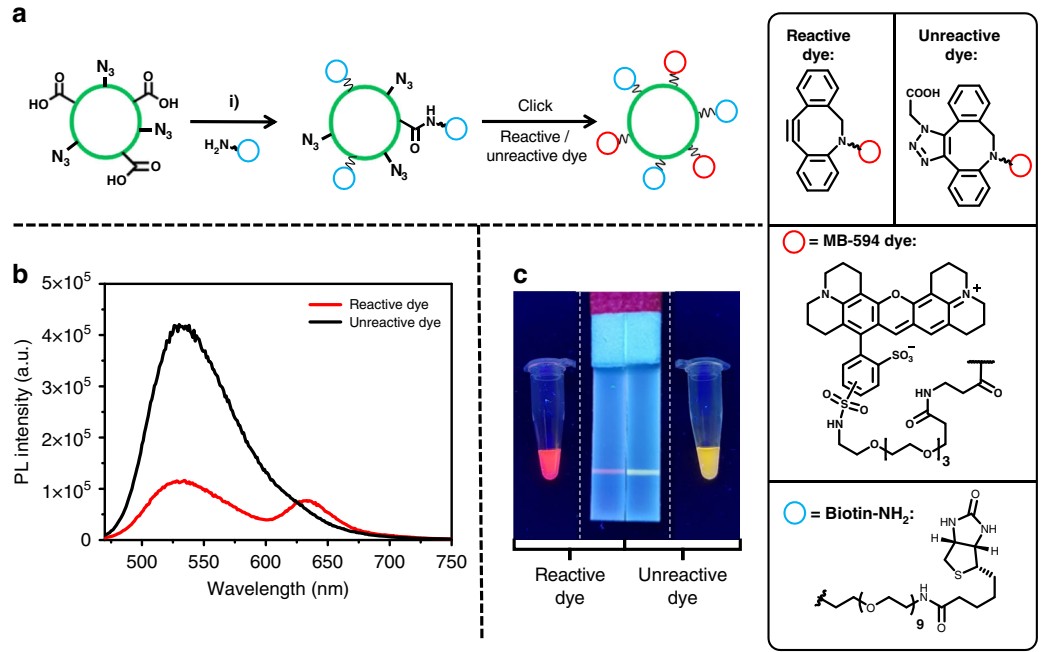

**Fig. 6** SPN surface bifunctionalisation. **a** Schematic of the modification of the surface of SPN-multi-nanoparticles with biotin followed by DBCO-594, (i) 1-ethyl-3-(3-dimethylaminopropyl)carbodiimide (EDC) and N-hydroxysulfosuccinimide (sulfo-NHS). **b** PL spectra of the nanoparticle solutions with reactive and unreactive dye (red and black lines, respectively). Samples were pulsed at 450 nm, with 10 vol% THF added. **c** Images of lateral flow strips run with unreactive and reactive dye nanoparticle solutions alongside the solutions (with 10 vol% THF added) all irradiated with UV light (365 nm)

achieved on the strip by reacting DBCO dye directly on the biotin functionalised **SPN-Multi** particles bound to the test line (Supplementary Figure 40). These tests demonstrate bifunctionalisation of a class of easily modifiable SPNs using orthogonal bioconjugated reactions, something previously not reported for SPNs and very challenging to achieve using previous polymer and particle functionalisation techniques.

## Discussion

Nucleophilic fluoride displacement is a powerful technique to readily functionalise conjugated polymer backbones with a variety of functional groups containing sidechains under highly controlled conditions. Unlike exisiting protocols which typically result from copolymerisation techniques, we can readily modify a single batch of consistent molecular weight polymer. This powerful tool enables researchers to rapidly develop structure–activity relationships for these polymers and introduce a wide range of functional groups. We exemplified the utility of this approach by preparing a range of emissive semiconductor polymer nanoparticles (SPNs) with controlled surface chemistry and demonstrated that the multiple different reactive groups introduced are present and retain reactivity at the surface of these particles.

In the realm of SPNs, our approach will enable advanced combinations of functionalities. For example, we can envision multi-targeted and multi-mode systems where both cell-penetrating peptides[19] and antibodies[34] or nucleic acids[40] are presented on particles that are capable of NIR-fluorescent[41–43] and magnetic resonance reporting[44,45], controlled drug release[46], and/or photothermal actuation[45,47]. Furthermore, we envisage significant impact in the wider field of semiconducting polymers, where our method will enable ready access to a range of chemical functionality scarcely accessible before.

## Methods

**Materials**. All solvents and chemicals were purchased from Sigma-Aldrich unless stated otherwise. All reactions were carried out under an inert argon atmosphere

using standard Schlenk line techniques, with dry solvents, unless stated otherwise. Active dye (DBCO-MB 594) was purchased from Kerafast. MilliQ water was used throughout the nanoparticle studies. Details of the supply or synthesis of monomers can be found in Supplementary Methods.

**Instruments and measurements**. A Biotage initiator V 2.3., in constant temperature mode, was used for all microwave reactions. NMR spectra were recorded on Bruker AV-400 (400 MHz) spectrometer. Weight-average ($M_w$) and Number-average ($M_n$) molecular weights were determined with an Agilent-Technologies 1260 Infinity GPC System with 1260 RID and DAD VL attachments. Measurements were performed at 80 °C, using analytical grade chlorobenzene as eluent with two PLgel 10 μm MIXED B columns in series. Molar mass, as a function of elution time through the columns was calibrated using known molecular weight Agilent EasiVial narrow dispersity polystyrene standards. Samples were prepared using analytical grade chlorobenzene in concentrations of ~1–2 mg mL$^{-1}$ and filtered with VWR PES membrane 0.45 μm syringe filters before submission. An injection volume of 50 μL and GPC flow rate of 1.00 mL min$^{-1}$ was used. GPC analysis of **P2-Silane** was not run due to potential crosslinking during measurement, which could damage the equipment. **P2-Copolymer** did not pass through GPC column so a molecular weight was not obtained.

A Shimadzu UV-1800 UV–Vis spectrophotometer was used to measure UV–Vis absorption. PL spectra were acquired on a Fluorolog® FL3-31 spectrofluorometer using a Hellma® lltra micro-fluorescence cuvette. Dynamic light scattering (DLS) were performed using a ZetaSizer Nano ZS (Malvern). Nanoparticles tracking analysis (NTA) was acquired on a Malvern Instruments NS3000. Solutions were diluted 100× with water before injection. The capture settings of the NTA were kept consistent for all samples at a screen gain of 1 and a camera level of 11 and the process settings were kept consistent at a screen gain of 11 and a detection threshold of 5. Samples were prepared for scanning transmission electron microscopy (STEM) by drop casting onto TEM grids with holey carbon support films. The imaging and energy-dispersive X-ray spectoscopy (EDX) was carried out in scanning TEM (STEM) mode using a JEM-2100F microscope at 200 kV. Polystreptavidin test strips were purchased from Mologic and photographs were acquired on the camera of a Samsung S8 mobile phone whilst the strips were illuminated by two 6 W black light bulbs (BLB-T5/6 W) in a dark room. Time-correlated single photo counting (TCSPC) setup was acquired using a DeltaFlex (Horiba) system, exciting with pulsed 467 nm light and fluorescence was detected at 535 nm (SPC-650 detector, Horiba) with a 515 nm longpass filter. PLQY was measured using a Jobin Yvon Horiba Fluoromax-4 spectrofluorometer equipped with a Quanta-φ integrating sphere attachment (excitation wavelength $\lambda_{ex} = 450$ nm). Self-absorption was corrected using method reported by de Mello et al.[48] Attenuated total reflectance infrared (ATR-IR) spectroscopy was recorded on a Perkin-Elmer spectrum BX.

MALDI-TOF measurements were performed with a Waters MALDI micro-MX spectrometer in linear positive mode on a stainless steel plate. 1 μL of a solution of trans-2-[3-(4-tert-Butylphenyl)−2-methyl-2-propenylidene]malononitrile (DCTB) dissolved in THF (10 mg/mL) with polymer added (1 mg) was added to the steel plate. Once dry, the sample was measured using ca. 60% laser intensity. The resulting spectra were averaged using the built-in functions from the Masslynx software. Each spectrum was then normalised against the total ion count (TIC). Samples were referenced against a sample of adrenocorticotropic hormone (ACTH) with a known mass of 2466.1 (ACTH + H+). Signals corresponding to species of increasing oligomer length were observed (Supplementary Figure 5). The error in absolute values was a result of the high signal to noise which affected the assignment of the apex of the peaks.

Bottom-contact bottom-gate (BC-BG) organic thin film transistors (OTFTs) were fabricated to study the charge carrier mobility of the **P1-SR-Suz** and **P1-SR** polymers. Heavily p-doped Si wafers, acting as the common gate electrode, incorporating with a thermally oxidised 200 nm-thick $SiO_2$ layer as the gate dielectric, were employed. Bilayer source–drain electrodes composed of Ti/Au were patterned via photolithography forming channel widths of 1000 μm and lengths in the range of 1.25–40 μm. The surface of the $SiO_2$ dielectric layer was treated with the primer hexamethyldisilazane prior to polymer deposition to minimise the interaction between the semiconductor and the dielectric. Polymer semiconductor solutions (10 mg ml$^{-1}$ in chlorobenzene) were deposited via spin coating at 2000 rpm for 1 min in nitrogen. Electrical characterisation of the transistors was conducted in nitrogen glovebox using an Agilent B2902A semiconductor.

**Polymer synthesis**. Synthesis of precursor polymers **P1–P6** can be found in Supplementary Methods.

**P1-SR, P2-Copolymer, P2-Silane, P3-SR, P4-SR, P5-SR** and **P6-SR**: Detailed below is a general protocol for the reaction of polymers **P1–P6** with thiols. Details of the synthesis and characterisation of individual polymers can be found in Supplementary Methods.

Polymer and a large excess of $K_2CO_3$ were added to a high-pressure microwave vial. The vial was sealed with a septum and degassed with argon, before anhydrous chlorobenzene and DMF (3:1 v:v at a concentration of 5 mg/mL of polymer) were added. Excess thiol was then added and the solution was degassed with argon. The solution was heated in a microwave reactor at 120 °C for 30 min. After cooling, the solution was precipitated into methanol (10:1 v:v of methanol:solvent), stirred for 30 min and filtered through a Soxhlet thimble. Unreacted thiol was removed by washing (Soxhlet) with acetone and the polymer was extracted with $CHCl_3$. The $CHCl_3$ fraction was concentrated to 1 mL, precipitated into MeOH and filtered to isolate the polymer.

Substitution of **P2-F** with increasing thiol content: **P2-F** (5 mg on average, 9.24 μmol (variation for each reaction, see Supplementary Methods)) and $K_2CO_3$ (10 mg, 0.072 mmol) were added to a 2 mL high-pressure microwave vial. The vial was sealed with a septum and degassed with argon, before anhydrous chlorobenzene (0.75 mL) and DMF (0.25 mL) were added. Dodecanethiol solution (varied amount, see Supplementary Methods) was then added and the solution was degassed with argon. The solution was heated in a microwave reactor at 150 °C for 60 min. After cooling the solution was precipitated into methanol (10 mL), stirred for 30 min and filtered through a Soxhlet thimble. Unreacted thiol and DMF was removed by washing (Soxhlet) with methanol and the polymer was extracted with $CHCl_3$. The $CHCl_3$ was removed under reduced pressure, resulting in a yellow solid. $^1$H NMR (400 MHz, CDCl$_3$) δ 8.07–7.60 (m, 7 H), 3.02–2.93 (m, varied (see Supplementary Methods), 2.24–1.93 (m, 4 H), 1.42–1.09 (m, varied), 0.92–0.73 (m, 9 H).

**P2-N$_3$, P2-Alkene, P2-OR** and **P2-PEG**: Detailed below is a general protocol for the reaction of **P2-F** with thioacetates and alcohols. Details of the synthesis and characterisation of individual polymers can be found in Supplementary Methods.

**P2-F** was added to a high-pressure microwave vial. Chlorobenzene and DMF were added (3:1 v:v at a concentration of 5 mg/mL of polymer) under ambient atmosphere and the solution was heated to 100 °C. When the polymer was fully dissolved a pellet of potassium hydroxide was added, and the solution stirred for 10 min before adding the appropriate thioacetate or alcohol. The solution was heated for a further 10 min, in the dark. The resulting solution was allowed to cool to room temperature before the addition of $CHCl_3$. The organics were washed with water, concentrated to 3 mL and precipitated into methanol and stirred for 30 min. Unreacted nucleophile was removed by washing (Soxhlet) with acetone and the polymer was extracted into $CHCl_3$. The $CHCl_3$ fraction was concentrated to 1 mL, precipitated into methanol (10:1 v:v of methanol:solvent) and filtered, resulting in a yellow solid.

**P2-COOH**: **P2-F** (13.4 mg, 0.025 mmol, batch 1), 11-mercaptoundecanoic acid (1.7 mg, 7.78 μmol) and $K_2CO_3$ (100 mg, 0.72 mmol) were added to a 5 mL high-pressure microwave vial. The vial was sealed with a septum and degassed with argon, before anhydrous chlorobenzene (2.25 mL) and DMF (0.75 mL) were added. The solution was heated in a microwave reactor at 150 °C for 1 h. The solution was allowed to cool to room temperature before a few drops of acetic acid was added. The solution was then diluted with $CHCl_3$ and washed with saturated aqueous ammonium chloride (20 mL) followed by water (20 mL). The solution was concentrated to 3 mL under reduced pressure and precipitated into methanol (10 mL), stirred for 30 min and filtered. The polymer was washed with acetone and hexane to remove disulphide impurity and dried to afford a yellow solid (12.9 mg,

86%). $M_n$: 27.4 kDa, $M_w$: 57.1 kDa, $M_w/M_n$ (Đ): 2.08. $^1$H NMR (400 MHz, CDCl$_3$) δ 8.18–7.59 (m, 7H), 3.06–2.89 (br, 0.5H), 2.43–1.96 (m, 4H), 1.37–0.89 (m, 30H), 0.87–0.79 (m, 6H). IR 1704 cm$^{-1}$ (C = O).

**P2-SAc**: **P2-F** (5.8 mg, 0.011 mmol, batch 1), 1,8-octanedithiol diacetate (30 mg, 0.130 mmol) and a stirrer bar were added to a 2 mL high-pressure microwave vial. Chlorobenzene (0.75 mL) and DMF (0.25 mL) were added and the solution was heated to 100 °C under ambient conditions. NaOH in methanol (20 μL of 188 mM, 3.8 μmol) was added and the solution was heated for 10 min. A further 20 μL was then added and the solution was heated for a further 10 min. The resulting solution was precipitated into methanol (20 mL) and stirred for 30 min. Unreacted thioacetate was removed by washing (Soxhlet) with acetone and the polymer was extracted with $CHCl_3$. The $CHCl_3$ fraction was concentrated to 1 mL and precipitated into methanol and filtered, resulting in a yellow solid (3 mg, 45%). $M_n$: 34.6 kDa, $M_w$: 52.3 kDa, $M_w/M_n$ (Đ): 1.51. $^1$H NMR (400 MHz, CDCl$_3$) δ 8.18–7.60 (m, 7H), 3.02–2.90 (m, 0.7H), 2.84 (t, $J$ = 7.4 Hz, 0.70H), 2.31 (s, 1H), 2.27–1.97 (m, 2H), 1.32–1.00 (m, 30H), 0.85–0.73 (m, 6H); IR 1750 cm$^{-1}$ (C=O).

**P2-Multi** (synthesised over two steps): 25 mol% of 11-mercaptoundecanoic acid was reacted with **P2-F** (batch 4) following the method used to synthesis **P2-COOH**. This yielded 40 mg of a polymer with 20 mol% carboxylic acid groups on the backbone (estimated from NMR analysis). The resulting polymer (20 mg, 0.034 mmol) and a stirrer bar were added to a 5 mL high-pressure microwave vial. Chlorobenzene (3 mL) and DMF (1 mL) were added under ambient atmosphere and the solution was heated to 100 °C, when the polymer was fully dissolved a pellet of potassium hydroxide was then added. The solution was stirred for 10 min before adding triethylene glycol monomethyl ether (3.4 mg, 0.021 mmol). The solution was heated for a further 10 min (an aliquot (10 μL) of reaction mixture was diluted with $CHCl_3$ (3 mL) for UV–Vis analysis). S-(3-azidopropyl)thioacetate (10 μL, 0.07 mmol) was then added to the reaction mixture and the reaction was heated for an additional 10 min, in the dark. The resulting solution was allowed to cool to room temperature before the addition of 5 mL $CHCl_3$. The organics were washed with water (2 × 30 mL), concentrated to 3 mL and precipitated into methanol (20 mL), filtered and washed with hot MeOH (3 × 20 mL). The polymer was dried under vacuum, resulting in a yellow solid (20.3 mg, 86%). $M_n$: 40.1 kDa, $M_w$: 65.6 kDa, $M_w/M_n$ (Đ): 1.64. $^1$H NMR (400 MHz, CDCl$_3$) δ 8.11–7.56 (m, 7H), 4.35 (br, 1.2H), 3.81 (br, 1.3H), 3.70–3.57 (m, 3.7H), 5.13 (br, 1.3H), 3.40–3.28 (m, 2.1H), 3.06–2.93 (m, 0.7H), 2.20–1.81 (m, 4H), 1.36–0.59 (m, 34H).

**P2-Multi** (synthesised in one-pot): **P2-F** (19 mg, 0.035 mmol), 11-mercaptoundecanoic acid (1.7 mg, 7.78 μmol) and KOH (one pellet) were added to a 5 mL high-pressure microwave vial. The vial was sealed with a septum and degassed with argon, before anhydrous chlorobenzene (3 mL) and DMF (1 mL) were added. The solution was heated in a microwave reactor at 150 °C for 1 h. The solution was allowed to cool to room temperature, the septum was removed and an aliquot (10 μL) of reaction mixture was then added to $CHCl_3$ (3 mL) for UV–Vis analysis. The reaction mixture was heated (in an oil bath) to 100 °C a further pellet of KOH was then added. The solution was stirred for 10 min before adding triethylene glycol monomethyl ether (3.0 μL, 0.017 mmol). The solution was heated for a further 10 min (an aliquot (10 μL) of reaction mixture was diluted with $CHCl_3$ (3 mL) for UV–Vis analysis). S-(3-azidopropyl)thioacetate (20 μL, 0.14 mmol) was then added and the reaction heated for an additional 10 min, in the dark. The resulting solution was allowed to cool to room temperature before the addition of 5 mL $CHCl_3$. The organics were washed with water (2 × 30 mL), concentrated to 3 mL and precipitated into methanol (20 mL), filtered and washed with hot MeOH (3 × 20 mL). The polymer was dried under vacuum, resulting in a yellow solid (18 mg, 74%). $^1$H NMR (400 MHz, CDCl$_3$) δ 8.11–7.60 (m, 7H), 4.35 (br, 1.4H), 3.81 (br, 1.4H), 3.70–3.57 (m, 4.2H), 5.13 (br, 1.4H), 3.40–3.28 (m, 2.7H), 3.07–2.91 (m, 0.6H), 2.23–1.83 (m, 4H), 1.36–0.70 (m, 34H).

Nanoparticle fabrication: Polymer solutions in THF (0.1 mg/mL) were prepared from **P2-N$_3$, P2-COOH, P2-Silane** and **P7-Multi** polymers. Each solution was filtered through a 0.2 μm PTFE filter. The solution (1 mL) was then injected rapidly into 9 mL of water under sonication and left under sonication for 3 min. The resulting nanoparticle suspensions were heated to 60 °C and bubbled with $N_2$ for 1 h to remove THF. Resulting solutions were then filtered through a 0.2 μm cellulose filter to remove large particulates.

Click of DBCO-dye to **SPN-N$_3$**: Concentrations of **SPN-N$_3$** were found to be $3.4 \times 10^{11}$ particles/mL (0.56 nM) using NTA. To prepare the unreactive dye solution, 10 μL of DBCO-MB 594 (500 μM, in DMSO) was reacted with azidoacetic acid (1 μL, 0.013 mmol) at 4 °C in the dark in for 4 h. This resulted in a 454.6 μM solution of unreactive dye. Reactive dye solutions were made up to same concentration by adding 1 μL DMSO to 10 μL of DBCO-MB 594 (500 μM, in DMSO). These solutions were both then diluted to a concentration of 1 μM by adding 1 μL of dye DMSO solution to 453.55 μL of water. To solutions of **SPN-N$_3$** (10 μL, 5.6 fmol) was added an increasing quantity of reactive dye solutions (see Supplementary Table 2). Solutions were left at 4 °C for 18 h in the dark. Each solution was then made up to 100 μL with water. The PL spectra and TCSPC of each solution was then recorded.

EDC coupling of biotin-NH$_2$ to **SPN-COOH**: 1 μg/mL solutions of N-(3-dimethylaminopropyl)-N′-ethylcarbodiimide hydrochloride (EDC) and N-hydroxysulfosuccinimide (Sulfo-NHS) were made up in water. O-(2-Aminoethyl)-O′-[2-(biotinylamino)ethyl]octaethylene glycol (biotin-NH$_2$) solutions were made up to 1, 10, 100 and 200 μg/mL in water. To four solutions of **SPN-COOH** (100 μL) was added EDC solution (50 μL, 0.26 nmol) and Sulfo-NHS solution

(50 μL, 0.23 nmol). These solutions were left to react at room temperature for 15 min. The four different solutions of biotin-NH$_2$ (15 μL) was then added to each reaction mixture. The mixtures were then left to react for 48 h at room temperature whilst shaking. To a further four solutions of **SPN-COOH** (100 μL) was added Sulfo-NHS solution (50 μL) and water (50 μL) and were treated in the same way to act as controls for non-specific binding. 10 μL of each resulting solution was added to a mixture of foetal bovine serum (FBS) and TWEEN®20 (0.02 vol%) (50 μL) and incubated for 5 min. To each mixture, poly-streptavidin test strips were added so the solution which was allowed to run up the test strip under capillary action and then left until dry. Strips were illuminated under UV light (365 nm) and photographs taken.

EDC coupling of biotin-NH$_2$ to **SPN-Multi**: 1 μg/mL solutions of *N*-(3-dimethylaminopropyl)-*N*′-ethylcarbodiimide hydrochloride (EDC) and *N*-hydroxysulfosuccinimide (Sulfo-NHS) were made up in water. *O*-(2-aminoethyl)-*O*′-[2-(biotinylamino)ethyl]octaethylene glycol (biotin-NH$_2$) solutions were made up to 1, 10, 100 and 200 μg/mL in water. To four solutions of **SPN-Multi** (100 μL) was added EDC solution (50 μL, 0.26 nmol) and Sulfo-NHS solution (50 μL, 0.23 nmol). These solutions were left to react at room temperature for 15 min. The four different concentrations of biotin-NH$_2$ (15 μL) was then added to each solution. The mixtures were then left to react for 48 h at room temperature whilst shaking. To a further four solutions of **SPN-Multi** was added Sulfo-NHS solution (50 μL) and water (50 μL) and were treated in the same way to act as controls for non-specific binding. 10 μL of each solution was added to a mixture of foetal bovine serum (FBS) and TWEEN®20 (0.02 vol%) (50 μL) and incubated for 5 min. To each mixture, poly-streptavidin test strips were added, so the solution which was allowed to run up the test strip under capillary action and then left until dry. Strips were illuminated under UV light (365 nm) and photographs taken.

Click of DBCO-dye to **SPN-multi**: To 12.9 pM solutions (calculated using NTA, and then equalised using water) of **SPN-Multi** and **SPN-COOH** (50 μL, 0.6 fmol) was added 0.2 μL of reactive dye (454.6 μM, in DMSO) or 0.2 μL of unreactive dye (454.6 μM, in DMSO) solutions (see above for method of making dye solutions). Solutions were left to react for 18 h at 4 °C. Reaction mixtures were then made up to 100 μL (with 10 vol% THF) and the PL spectra were recorded.

Tandem EDC coupling and DBCO click to **SPN-Multi**: **SPN-Multi** were reacted with biotin-NH$_2$ (15 μL of 10 μg/mL) using the method described above, to yield a biotin-functionalised nanoparticle solution. To aliquots of this solution (49.8 μL) was added reactive dye solution or unreactive dye solution (454.6 μM, 0.25 μL). Solutions were left to react for 18 h at 4 °C. The solutions were irradiated under UV-light (365 nm) and photographs taken. 10 μL of each solution was added to a mixture of foetal bovine serum (FBS) and TWEEN®20 (0.02 vol%) (50 μL) and incubated for 5 min. To each mixture, poly-streptavidin test strips were added so the solution which was allowed to run up the test strip under capillary action and then left until dry. Strips were illuminated under UV light (365 nm) and photographs taken. The remaining reaction solutions were made up to 100 μL (with 10 vol% THF) and the PL spectra was recorded.

**Data availability**. All data relating to this publication can be found at https://doi.org/10.5281/zenodo.1284917.

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

## Acknowledgements

M.H. acknowledges financial support from the EPSRC (grant numbers: EP/G037515/1; EP/L016702/1) and the British Council (337323). We thank CSIRO for the sponsorship of A.C., and the China Scholarship Council for S.C. M.M.S. acknowledges support from the ERC Seventh Framework Programme Consolidator grant "Naturale CG" (616417), the Wellcome Trust Senior Investigator Award (098411/Z/12/Z), the Imperial College BHF Centre for Cardiac Regeneration (RM/13/1/30157), the i-sense EPSRC IRC in Early Warning Sensing Systems for Infectious Diseases (EP/K031953/1) and the EPSRC grant "Bio-functionalised nanomaterials for ultrasensitive biosensing" (EP/K020641/1).

## Author contributions

A.C. synthesised polymers, nanoparticles, analysed data and wrote the manuscript. C.S. W. synthesised nanoparticles, analysed data and wrote the manuscript with A.C. A.C. (Casey) gave input on polymer synthesis methods and synthesised polymers **P5-F**, **P5-SR** and **P3-F**. S.C. synthesised polymers **P2-OR**, **P2-PEG** and **P2-Copolymer** and two batches of **P2-F**. A.V.M. analysed the polymer characterisation data. T.W. ran the MALDI-TOF experiments. I.H. calculated the PLQYs. C.H.B. and M.A.M. ran the STEM experiments and analysed the data. J.P. fabricated the OTFTs and analysed the data with the assistance of T.A. P.D.H. gave extensive input on the study process and manuscript. R.G. calculated the FRET efficiencies from the PL and TCSPC data. Z.P. helped synthesise **P5-F**. M.M.S. and M.H. conceived and/or designed experiments, directed the work and edited the paper. All authors contributed to the discussion.

## Additional information

**Competing interests:** The authors declare no competing interests.

