## [Peer Review File · Nature Communications]

Reviewers' comments:

Reviewer #1 (Remarks to the Author):

The manuscript of Creamer et al. gives an interesting contribution in the field of functionalization of conjugated polymers by post-polymerization functionalization, which is barely explored. Particularly, the authors address the issue of how introducing single as well as multiple functionalities into a polymer backbone and, ultimately, polymer nanoparticles in a controlled manner.

Overall, the manuscript provides a very informative picture of the versatility of the proposed novel, selective, high-yielding post-polymerization functionalization protocol. The data are in good order, clearly discussed, and well-support the conclusions.

A few points should be addressed before publication:

- page 2, Introduction, text line 8: "Therefore is if often" should be "Therefore it is often"
- page 4, Results and discussion, text line 3: "[...]than the carbazole copolymer (Mn= [...])" should be "[...]than the carbazole copolymer P1-F (Mn= [...])", for clarity
- page 5, Figure 1a: the label "P2-F" should be added below the polymer structure
- page 6, text line 1: "It should be noted that upon the formation of polymer nanoparticles in water, the substitution reaction had little impact on the PLQY". The authors did not introduce nanoparticles yet, at this point, and the sentence can therefore be confusing. I would move this observation when discussing nanoparticles PL properties or, at least, the sentence should be modified into "It should be noted that upon the formation of polymer nanoparticles (vide infra) in water, the substitution reaction had little impact on the PLQY"

Reviewer #2 (Remarks to the Author):

This is a competently carried out piece of work, but when one looks at the overall impact, it is unclear what innovative method has been achieved. There is no novel synthetic methodology developed here – the nanoparticles are prepared in exactly the same route as reported numerous times before. Whilst the claim that this is post-polymerisation functionalisation of a conjugated polymer backbone is strictly true, it is far too close to the work of Moon et al. who carries out essentially the same methodology using altered PPE polymers. (Chem. Commun., 2007, 4910). Although this is obviously slightly different, the essence of the work – (modifying polymers to incorporate the functional group) is identical. Moon has also modified the backbone of polymers previously to introduce differently practical applications of conjugated polymer nanoparticles (Chem. Commun., 2016, 52, 4910). It is not clear to me what benefit this methodology contributes beyond a natural progression of existing work. Even the interesting functional chemistry isn't that novel - Functionalisation of the particles with the click group (N3) has been done in a simpler method (Chem Asian J. 2014 Nov;9(11):3121-4.). Whilst the work will undoubtedly be published somewhere, it is unlikely to make a major impact in the area of conjugated polymer nanoparticles. I would suggest an ACS journal such as Chemistry of Materials.

Reviewer #3 (Remarks to the Author):

This manuscript describes a post-polymerization modification method for introducing covalent functionality to conjugated polymers and polymer-based nanoparticles. The authors use nucleophilic aromatic substitution based on sulfur or oxygen nucleophiles to displace fluorine. Several different polymer backbones are used to demonstrate substrate scope and a broad variety of nucleophiles are used as well. It is clear that this method has potential great value and opens a pathway to a variety of polymer structures that would be inaccessible by existing methods. The multi-functional polymer structure is of special interest. The authors also demonstrate the method

on functional polymer nanoparticles, where additional functionality is provided by click reactions or amide bond forming reactions. The potential of the method for nanoparticle bio-applications is clear.

Overall, this is a well-written manuscript presenting a thorough and well-executed study. The novelty and impact is suitable for Nature Communications and this paper is recommended for publication.

One point that was omitted was a characterization of how the reaction might affect other electronic properties in the polymer, namely charge transport. While this is irrelevant to the nanoparticle application, the measurement of hole mobility in a direct comparison (e.g. P1-SR-Suz vs. P1-SR) would give some insight into other potential effects of the post-polymerization reaction on polymer structure and performance. This would be a nice experiment and would add to an already strong contribution.

Reply to Reviewers comments

Reviewer #1 (Remarks to the Author):

The manuscript of Creamer et al. gives an interesting contribution in the field of functionalization of conjugated polymers by post-polymerization functionalization, which is barely explored. Particularly, the authors address the issue of how introducing single as well as multiple functionalities into a polymer backbone and, ultimately, polymer nanoparticles in a controlled manner. Overall, the manuscript provides a very informative picture of the versatility of the proposed novel, selective, high-yielding post-polymerization functionalization protocol. The data are in good order, clearly discussed, and well-support the conclusions.

A few points should be addressed before publication :

We thank the referee for their positive review highlighting the novelty of our approach.

-page 2, Introduction, text line 8: “Therefore is if often” should be “Therefore it is often”

Changed

-page 4, Results and discussion, text line 3: “[...]than the carbazole copolymer (Mn= [...])” should be “[...]than the carbazole copolymer P1-F (Mn= [...])”, for clarity

Changed

-page 5, Figure 1a: the label “P2-F” should be added below the polymer structure

Changed

-page 6, text line 1: “It should be noted that upon the formation of polymer nanoparticles in water, the substitution reaction had little impact on the PLQY”. The authors did not introduce nanoparticles yet, at this point, and the sentence can therefore be confusing. I would move this observation when discussing nanoparticles PL properties or, at least, the sentence should be modified into “It should be noted that upon the formation of polymer nanoparticles (*vide infra*) in water, the substitution reaction had little impact on the PLQY”

As suggested we have added ‘vide infra’ to make it clear we will return to this discussion later.

Reviewer #2 (Remarks to the Author):

This is a competently carried out piece of work, but when one looks at the overall impact, its unclear what innovative method has been achieved. There is no novel synthetic methodology developed here – the nanoparticles are prepared in exactly the same route as reported numerous times before. Whilst the claim that this is post-polymerisation functionalisation of a conjugated polymer backbone is strictly true, its far too close to the work of Moon et al. who carries out essential the same methodology using altered PPE polymers. (Chem. Commun., 2007, 4910). Although this is obviously slightly different, the essence of the work – (modifying polymers to incorporate the functional group) is identical. Moon has also modified the backbone of polymers previously to introduce differently practical applications of conjugated polymer nanoparticles (Chem. Commun., 2016, 52, 4910). It is not clear to me what benefit this methodology contributes beyond a natural progression of exiting work. Even the interesting functional chemistry isn’t that novel - Functionalisation of the particles

with the click group (N3) has been done in a simpler method (Chem Asian J. 2014 Nov;9(11):3121-4.). Whilst the work will undoubtedly be published somewhere, it is unlikely to make a major impact in the area of conjugated polymer nanoparticles. I would suggest an ACS journal such as Chemistry of Materials.

The major difference of our approach to the references mentioned is that we directly modify the conjugated backbone, rather than just the alkyl sidechains, as in all the referenced work. Therefore our approach modifies the fundamental polymer properties such as band gap and solubility (see for example figure 1 and 2 and supplementary figure 14, and corresponding discussion). The previous literature does not report this. Our approach also allows the ready incorporation of multiple functional groups, something very rarely reported previously. For example we can find no examples of conjugated PEG containing polymers functionalised with azide and carboxylic acid groups (See discussion on page 8). We agree that these points on the novelty could have been articulated more clearly and so have now updated (page 2/3) as follows.:

Here we report a novel and widely applicable strategy to directly modify the conjugated backbone via a quantitative nucleophilic aromatic substitution (S_NAr) reaction on fluorinated electron deficient comonomers within the polymer backbone (Scheme 1a). Such functionalization results in significant changes to the optoelectronic and physical properties of the polymer. It also enables the facile introduction of reactive handles for further functionalization or cross-linking. Multiple reactive groups can be introduced in a controlled one-pot reaction, something current co-polymerisation approaches are very rarely reported to achieve¹⁵. This is of particular interest in the field of semiconducting polymer nanoparticles (SPNs) which are relatively challenging to functionalise. Existing strategies to SPN functionalisation include encapsulation by an inorganic shell and subsequent chemical modification¹⁶⁻¹⁸, or coprecipitation/emulsification with amphiphilic molecules containing reactive groups^{19,20,21}. This later approach relies on non-covalent interactions between the polymer and the functionalised counterpart, which can dissociate over time²², particularly at elevated temperature. All of these approaches also rely on dilution of the emissive component, which inherently limits the brightness of the particule for a given size. Covalent attachment of functional groups directly to the semiconducting polymer backbone overcomes these problems²³, but has been synthetically challenging, particularly if multiple functional groups are required. Our new approach to synthesising these materials enables the facile creation of functionalised polymers and particles with multiple different reactive handles covalently linked on their surface.

Both reviewer 1 (.....polymers by post-polymerization functionalization... which is barely explored...; ...versatility of the proposed novel, selective,...) and reviewer 3 (...potential great value and opens a pathway to a variety of polymer structures that would be inaccessible by existing methods....; potential of the method for nanoparticle bio-applications is clear.) highlighted the novelty of the methodology.

Reviewer #3 (Remarks to the Author):

This manuscript describes a post-polymerization modification method for introducing covalent functionality to conjugated polymers and polymer-based nanoparticles. The authors use nucleophilic aromatic substitution based on sulfur or oxygen nucleophiles to displace fluorine. Several different polymer backbones are used to demonstrate substrate scope and a broad variety of nucleophiles are

used as well. It is clear that this method has potential great value and opens a pathway to a variety of polymer structures that would be inaccessible by existing methods. The multi-functional polymer structure is of special interest. The authors also demonstrate the method on functional polymer nanoparticles, where additional functionality is provided by click reactions or amide bond forming reactions. The potential of the method for nanoparticle bio-applications is clear.

Overall, this is a well-written manuscript presenting a thorough and well-executed study. The novelty and impact is suitable for Nature Communications and this paper is recommended for publication.

We thank the referee for their positive review highlighting the novelty, scope and utility of our approach.

One point that was omitted was a characterization of how the reaction might affect other electronic properties in the polymer, namely charge transport. While this is irrelevant to the nanoparticle application, the measurement of hole mobility in a direct comparison (e.g. P1-SR-Suz vs. P1-SR) would give some insight into other potential effects of the post-polymerization reaction on polymer structure and performance. This would be a nice experiment and would add to an already strong contribution.

We agree this is an interesting question that was not addressed in the first draft. We have now measured the performance of both P1-SR-Suz and P1-SR in thin-film transistor devices, and include discussion on their (comparable) performance as below (page 4 and supplementary figure 4 and table 1)

Given that the performance of semiconducting polymers in electronic device like organic thin-film transistors (OTFTs) is very sensitive to the presence of chemical defects or impurities, the performance of both polymers was also investigated in bottom gate, bottom contact OTFTs. Gratifyingly both polymers exhibited similar overall performance, with comparable charge carrier mobility (see Supplementary Information section, Fig. 4 and Table 1), providing further evidence that the displacement chemistry does not adversely affect polymer performance.

The additional TFT measurements required some additional expertise, and we have therefore added two more authors (Julianna Padini and her supervisor Prof Thomas Anthopolous). The author list and contributions have been updated, as well as the experimental section to include details of OTFT fabrication.

REVIEWERS' COMMENTS:

Reviewer #3 (Remarks to the Author):

The revised version of the manuscript has adequately addressed all concerns raised in the original version and is now recommended for publication.